# Immunoinformatics Approach to Design Multi-Epitope-Based Vaccine against Machupo Virus Taking Viral Nucleocapsid as a Potential Candidate

**DOI:** 10.3390/vaccines10101732

**Published:** 2022-10-17

**Authors:** Muhammad Naveed, Syeda Izma Makhdoom, Urooj Ali, Khizra Jabeen, Tariq Aziz, Ayaz Ali Khan, Sumbal Jamil, Muhammad Shahzad, Metab Alharbi, Abdulrahman Alshammari

**Affiliations:** 1Department of Biotechnology, Faculty of Science and Technology, University of Central Punjab, Lahore 54590, Pakistan; 2Department of Biotechnology, Quaid-I-Azam University Islamabad, Islamabad 45320, Pakistan; 3School of Food & Biological Engineering, Jiangsu University, Zhenjiang 212013, China; 4Department of Biotechnology, University of Malakand, Chakdara 18800, Pakistan; 5Rehman Medical Institute, Peshawar 25000, Pakistan; 6School of Biological Sciences, Health and Life Sciences Building, University of Reading, Reading RG6 6AX, UK; 7Department of Pharmacology and Toxicology, College of Pharmacy, King Saud University, P.O. Box 2455, Riyadh 11451, Saudi Arabia

**Keywords:** Machupo virus, Bolivian hemorrhage fever, B and T-cell epitopes, potential vaccine

## Abstract

The family members of Arenaviridae include members of the genus Machupo virus, which have bi-segmented negative sense RNA inside the envelope and can be transferred to humans through rodent carriers. Machupo virus, a member of the *mammarenavirus* genus, causes Bolivian hemorrhage fever, its viral nucleocapsid protein being a significant virulence factor. Currently, no treatment is available for Bolivian hemorrhage fever and work to develop a protective as well as post-diagnosis treatment is underway. Adding to these efforts, this study employed a reverse-vaccinology approach to design a vaccine with B and T-cell epitopes of the viral nucleocapsid protein of the Machupo virus. Five B-cell specific, eight MHC-I restricted, and 14 MHC-II restricted epitopes were finalized for the construct based on an antigenicity score of >0.5 and non-allergenicity as a key characteristic. The poly-histidine tag was used to construct an immunogenic and stable vaccine construct and 50S ribosomal 46 protein L7/L12 adjuvant with linkers (EAAAK, GPGPG, and AYY). It covers 99.99% of the world’s population, making it highly efficient. The physicochemical properties like the aliphatic index (118.31) and the GRAVY index (0.302) showed that the vaccine is easily soluble. The overall Ramachandran score of the construct was 90.7%, and the instability index was 35.13, endorsing a stable structure. The immune simulations demonstrated a long-lasting antibody response even after the excretion of the antigen from the body in the first 5 days of injection. The IgM + IgG titers were predicted to rise to 6000 10 days post-injection and were illustrated to be stable (around 3000) after a month, elucidating that the vaccine would be effective and provide enduring protection. Lastly, the molecular interaction between the construct and the IKBKE receptor was significant and a higher eigenfactor value in MD simulations confirmed the stable molecular interaction between the receptor and the vaccine, validating our construct.

## 1. Introduction

Viruses from the Arenaviridae family are encompassed, bi-portioned, negative-sense RNA infections that are transmitted to people via rat-related illnesses. The disease brought about by Arenavirus is not spread worldwide; instead, it is bound to common areas of the world. The arenavirus has a compact size, encapsulated in a lipid film whose hereditary material depends on RNA, and the family is segmented into different classes of virus-like: Biloi, Machupo, Lassa, Junin, Baskia, and Guanarito virus. These infections are characterized based on the disease they cause [1]. Machupo virus, a negative-sense mRNA virus from the Arenaviridae family, arose in 1952 in the rural areas of Bolivia. In 1962, 1000 individuals were affected by Machupo infection, and in 1963, an epidemic linked to the same virus emerged in Bolivia [2]. 

Starting in 1959 and proceeding through the mid-1960s, flare-ups of Bolivian hemorrhagic fever transpired in provincial regions all through Bolivia, along with the Beni division and the areas encompassing San Joaquin close to the eastern boundary of Bolivia [3]. Bolivians alluded to the baffling sickness as “Black Typhus,” implying the draining, high fever, agony, and quick demise brought about by the ailment. Following quite a while of examination, the U.S. government isolated Machupo from the Arenavirus family [4]. Karl Johnson from the NIH has grown a versatile laboratory glove box to reside in sterile settings to examine the infection in Bolivia. During the distinguishing proof cycle, various virologists got the disease, and almost 20% of San Joaquin locals died of the fever. Machupo has been remotely inflamed from then on, and the virus is endemic [5]. 

Aerosolized, food-borne, or direct contact with viral particles transmits the Machupo virus. These viral particles originate from the spitting, urinating, or defecating of the field mouse, Calomys callosus, a reservoir for the infection [6]. Today, it is known that flare-ups of Machupo occur when rich food sources or human living and cultivating patterns prompt an unanticipated expansion in the rat population [4]. The nursing and research center employees and the infected individual’s family have reported very few occurrences of individual-to-individual transfer of the virus [7]. The Machupo hemorrhagic fever is difficult to diagnose because it is like other endemic viral hemorrhagic fevers, such as Junin and Sabia, and many epidemic diseases, including dengue, malaria, and yellow fever [8] Prognosis may be expedited by helping doctors learn how to identify Bolivia hemorrhagic fever in a short time [9]. 

The Machupo virus contains two main segments, L and S, that infect bowls. The RNA-dependent polymerase is transcribed by the L segment and the Zinc finger by the Z segment. The nucleoprotein (N.P.) is transcribed by the S segment and the viral glycoprotein precursor GPC [10] and is known to have a critical role in the preliminary attachment of the viral surface to the host-cell receptors [11], making it a potential target for employing protective and combative strategies against Machupo. We know that the initial interaction with the host receptors is carried out by viral glycoproteins, and it is established that antibodies are generated against viral GPs, but the NP (nucleoprotein) has a significant role in protecting the viral RNA from degradation and no immunotherapy has been reported against it, making it a potential target for the current study. Furthermore, it is noted that the viral surface glycoproteins have historically been used in vaccine development. However, due to their considerable diversity, surface glycoproteins rarely cause long-lasting immunity [12]. On the other hand, NP is involved in a wide range of viral life cycle events, including coating and protecting viral RNA, controlling transcription and replication, and inducing host immunosuppression [12]. These shattered perspectives point to NP as a viable vaccination target, either as a standalone vaccine or as a complementing element, as has already been done with the Ebola vaccine design [12]. 

Vaccines have been considered substantial in preventing endemics and epidemics such as hemorrhagic fever, and the past decade has seen prodigious advancements in the development of computational vaccines using the reverse vaccinology approach [10,13,14]. Although immunotherapies like an attenuated vaccine (Candid#1) have been developed against Argentine hemorrhagic fever [15], their efficacy (84% for Candid#1), nature (live-attenuated), and limited target population (Argentina) advocate the need for a solution that is not limited and covers a broader population with more efficacy [16]. This study focuses on an effective vaccine design against the viral nucleocapsid of the Machupo virus using integrative computational and immunoinformatic approaches. It follows a multi-epitope-based vaccine design strategy with the evaluation of B- and T-cell epitopes, prediction of the population coverage, molecular interaction with the target host receptor, and elicitation of a stable immune response. As per our predictions, this investigation will assist researchers in developing a protective remedy against Machupo and hemorrhagic fever in the wake of future outbreaks. 

## 2. Materials and Methods

### 2.1. Sequence Retrieval

The nucleoprotein sequence of the Machupo virus used for the vaccine candidate was retrieved from the NCBI (https://www.ncbi.nlm.nih.gov/, accessed on 20 July 2022) [17].

### 2.2. Physiochemical Analysis

The physiochemical properties of the target protein candidate were analyzed through ExPasy ProtParam [18], available at https://www.expasy.org/resources/protparam (accessed on 20 July 2022). Molecular weight, half-life, and GRAVY were the characteristic properties estimated for target protein finalization. 

### 2.3. Allergenicity and Antigenicity Profiling of Proteins

Using VaxiJen 2.0 [19], available at http://www.ddg-pharmfac.net/vaxijen/VaxiJen/VaxiJen.html (accessed on 20 July 2022), and AllerTOP v2 [20], available at https://www.ddg-pharmfac.net/AllerTOP/method.html (accessed on 20 July 2022), we computed whether the protein was antigenic and non-allergenic to verify that the vaccine using its epitopes will be effective. 

### 2.4. Prediction of the Linear B-Cell Epitopes

The B-Cell Epitopes were computed by the Bepipred linear epitope prediction tool [21] available at the Immune Epitope Database, IEDB (http://tool.iedb.org/bcell/, accessed on 22 July 2022). The tool is based on the algorithm of the Hidden Markov Model, which determines the potential epitopes of a protein using a hidden amino acid propensity scale. BepiPred uses a hidden Markov model along with the propensity scale approach to estimate the location of linear B-cell epitopes [21].

### 2.5. Prediction of the MHC-Specific Epitopes

IEDB (Immune Epitope Database) was accessed at http://tools.iedb.org/mhci/ (accessed on 23 July 2022) and http://tools.iedb.org/mhcii/ (accessed on 23 July 2022) for the prediction of conserved domain-based cytotoxic and helper T cell epitopes, respectively. The ANN 4.0 algorithm [22] was used to predict MHC-I restricted epitopes based on its capacity to sort the results according to the IC_50_ values, while NN-align 2.3 [23] was used to predict the MHC-II restricted epitopes. All allele sets (HLA-A, -B, -C for MHC-I) and HLA-DRB, -DQB, and -DPB for MHCII) available on the IEDB database were used to evaluate the potential epitopes, and the epitope lengths were set at 9,10 for MHC-I and default for MHC-II.

### 2.6. Population Coverage

The population coverage tool [24] of the IEDB database, available at (http://tools.iedb.org/population/) was used on 25 July 2022 to evaluate the percentage of the world population targeted by our epitopes. Separate MHC-I, separate MHC-II, and combined modules were selected for the analysis, the population was set to be of the whole world, and a total of 20 epitopes, all 8 of MHC-I and 12 epitopes of MHC-II, with their target alleles, were taken as input.

### 2.7. Finalizing the Construct

Linkers (AAY, EAAAK, and GPGPG) are commonly employed in vaccine construction. The epitopes were fused together by linkers, and an adjuvant was added to stimulate the host immune response. Lastly, a poly-histidine tag (hexa-histidine tail) was added to the construct for purification assays. The VaxiJen 2.0 web server (http://www.ddg-pharmfac.net/vaxijen/VaxiJen/VaxiJen.html, accessed on 26 July 2022) [19] and AllerTop (https://www.ddg-pharmfac.net/AllerTOP/, accessed on 26 July 2022) [20] were used again to analyze the antigenicity and allergenicity of the construct. 

### 2.8. Physiochemical Properties of the Construct

The ExPasy of the ProtParam [18] (https://web.expasy.org/protparam/, accessed on 26 July 2022) and RaptorX [25] (http://raptorx.uchicago.edu/, accessed on 26 July 2022) were used to calculate the physicochemical properties and the secondary structure of the vaccine, respectively. The properties, including half-life of the vaccine, GRAVY index, and disordered structures, were computed to interpret the stability of the construct.

### 2.9. 3-D Structural Analysis of Vaccines and Receptors

The trRosetta [26] (https://yanglab.nankai.edu.cn/trRosetta/, accessed on 26 July 2022) was used to predict the 3D structure of the vaccine construct and the IKBKE receptor. The primary sequence of the receptor was retrieved from UniProtKB (https://uniprot.org, accessed on 27 July 2022) [27], having accession number Q14164 (IKKE_HUMAN). 

### 2.10. Refinement of the Construct and the Receptor

The structures of the vaccine and receptor were refined by the GalaxyRefine tool, available at http://galaxy.seoklab.org/ (accessed on 28 July 2022) [28]. Out of the five output models, the best was selected according to the most improved RAMA score.

### 2.11. Vaccine Construct Validation

The vaccine model was validated by the RAMPAGE server (https://www.ccp4.ac.uk/html/rampage.html, accessed on 28 July 2022) [29] and the Ramachandran plot was predicted. Every amino acid is plotted on the Ramachandran graph according to its Ψ value as a function of its Φ value. According to the plot, amino acids located in the top right section of the graph are most likely to form left-handed α-helices, whereas amino acids located on the bottom left indicate right-handed α helices. Amino acids in the top left section of the graph correspond to ß-sheets (parallel and anti-parallel and twisted) [29]. The amino acids plotted in the darker regions are the most favored ones, the ones in lighter regions are additionally allowed, and the ones located in the white regions are not allowed or are not of quality.

### 2.12. Molecular Docking with Host Receptor

The molecular docking was performed using the online server Cluspro2.0, available freely at https://cluspro.org/tut_dock.php, accessed on 30 July 2022 [30]. It is a protein–protein interaction software that computes interaction complexes according to interaction energies and centers. The docking complex was run on the online iMODs server, available at https://imods.iqfr.csic.es/, accessed on 31 July 2022 [31], to evaluate the stability of the docked complex. The eigen factor value was of significance here as it determined the energy required to deform the docking complex. A higher eigenvalue correlates to a stable docked complex. 

### 2.13. Expression Analysis

For the in-silico cloning of the vaccine candidate, an offline tool, SnapGene, was used retrieved from https://www.snapgene.com/, accessed on 2 August 2022 [32]. The pet28a plasmid was used to clone the vaccine construct to propagate the vaccine in *E. coli* and then in the human host.

### 2.14. Immune Stimulation

Using the C-ImmSim server, the immune response of the designed vaccine was simulated after cloning. It is available online at https://kraken.iac.rm.cnr.it/C-IMMSIM/, accessed on 3 August 2022 [33] and predicts the immune response like the natural response of the body. The methodology is summarized in Figure 1.

## 3. Results and Analysis

### 3.1. Sequence Retrieval

Under the specified accession number of CAA44486.1, the nucleocapsid protein of Machupo virus was retrieved from NCBI. The nucleocapsid protein has antigenic properties, which was confirmed by UniprotKB ID: P26578 (NCAP_MACHU). 

### 3.2. Physiochemical Analysis

Physiochemical properties were analyzed through ExPasy Protparam, through which the instability index was predicted at 42.31, the aliphatic index was 91.24, and the grand average of hydropathicity (GRAVY) was −0.449.

### 3.3. Allergenicity and Antigenicity Profiling of Nucleocapsid Protein

The allergenicity and antigenicity were predicted through the AllerTop 2.0 and VaxiJen online tools, respectively. AllerTop showed that the viral protein was probably non-allergenic with an antigenicity score of 0.558 (probable antigen) predicted by VaxiJen. 

### 3.4. B-Cell Epitope Prediction

The IEDB Linear Epitope Prediction Tool v2.0 was utilized to predict B-Cell Epitopes and identify the locations of the B-Cell Epitopes according to the Kolaskar and Tongaonkar antigenicity prediction algorithm. The maximum and minimum antigenicity were 1.4796 and 0.6472, respectively, with the average antigenicity being 1.015 as shown in Table 1. The protein’s antigenic determination threshold was 0.5, and all values greater than 0.5 were potential antigenic determinants. Five epitopes were found to be antigenic and, hence, were finalized.

### 3.5. T-Cell Epitope Recognition

The MHC-I and MHC-II restricted epitopes calculations returned thousands of data entries with IC50 values ranging from 1.28 to 2789.22, out of which the potential epitopes were filtered based on an IC_50_ value lower than 100. A low IC-50 value (100 is mostly used as a threshold) indicates that the vaccine epitope is effective at low doses and will thus exhibit less systemic toxicity when administered to an individual [34]. A lower IC_50_ value indicates that the epitopes used to construct the vaccine will generate an adequate immune response in a lower quantity of the vaccine [35].

#### 3.5.1. MHC-I-Restricting Epitopes

Four hundred and fifty MHC-I epitopes were predicted, out of which only eight were selected as target epitopes (Table 2) based on IC50 values lower than 100, antigenic potential, and non-allergenicity.

#### 3.5.2. MHC-II-Restricting Epitopes

Fourteen out of 586 MHC-II epitopes were selected based on their IC_50_ (Table 3), antigenicity, and allergenicity profiling. All the finalized epitopes were non-allergenic. The resulting table of MHC-II-specific epitopes provided two columns, one belonging to the core epitopes (AA length 9) and the other one with the expanded epitopes (variable lengths). The four epitopes with 9 AA lengths were selected because these core epitopes provided multiple expanded epitopes with lower IC50 values and similar allele restrictions. To avoid redundancy, we used the core epitopes instead of the expanded ones. The other epitopes, in Table 3, were chosen from the expanded epitopes’ column.

### 3.6. Population Coverage of the T-Cell-Specific Epitopes

All the alleles that restricted the finalized epitopes at an IC50 value under 100 were chosen for the population coverage analysis and are shown in Table 2 (MHC-I alleles) and Table 3 (MHC-II alleles). The population coverage computed by IEDB was 99.94% for the selected MHC-I epitopes, 87.31% for the selected MHC-II epitopes, and an excellent coverage of 99.99% for all the epitopes combined, as shown in Figure 2A. The individual Bolivian coverage (as the virus is currently endemic to Bolivia) was 84.24% (Figure 2B). The population coverage evidences that the epitopes chosen for this vaccine cover the world population and will be effective wherever utilized, with minor (non-significant) fluctuations in different ethnicities.

### 3.7. Vaccine Construct

Four components were added to the construct to make it stable, elicit a strong immune response, and be used for further purification assays. First, a 50S ribosomal 46 protein L7/L12 adjuvant with Uniprot ID: P9WHE3 was used. Secondly, linkers were used to fuse yet compartmentalize the B and T-cell-specific epitopes of the construct. The linkers used in the study were EAAAK, GPGPG, and AAY [36]. Protein folding, creating an extended conformation (flexibility), and separating functional domains all depend on linkers. Thirdly, epitopes were predicted based on allergenicity and antigenicity profiling. Lastly, the 6xHis tag was used for protein purification and identification. Adjuvant and B-cell epitopes were linked with the linker EAAAK, which increases protein folding, stability, and expression [36], whereas B-cell and MHC-I epitopes were linked with the linker GPGPG, aiding in flexibility [36], and MHC-I and MHC-II molecules were linked with the linker AAY to separate each functional epitope for MHC presentation [20], as elucidated in Figure 3. The antigenicity score of the vaccine construct was computed at 0.6628 at a threshold of 0.5. It was predicted to be non-allergenic, making it fit for further analysis. 

### 3.8. Secondary Structure and Physiochemical Properties of Vaccine

RaptorX indicated that there were 39% helices, 11% beta-sheets, and 48% coils in the construct. The solubility and accessibility of the vaccine construct predict that 43% of protein is exposed, 24% is in the trans-membrane region, and 31% is hydrophobic. The total amino acids of the vaccine construct were 438, along with a molecular weight of 47,171.30 kDa. The protein had an instability index (II) of 35.13, indicating that it is stable. The aliphatic score was 118.31, whereas the GRAVY (grand average of hydropathicity) was 0.302, predicting the construct is hydrophobic.

### 3.9. 3-D Structure of Vaccine and Receptor IKBKE

Figure 4 provides the tertiary structure of both the vaccine construct and the IKBKE receptor of the human predicted by trRosetta. This receptor was chosen for our study because it is an inhibitor of the nuclear factor, kappaB, and regulates inflammatory responses against viral infections, making it a decent target for the vaccine construct [37]. The confidence of model prediction for our vaccine was 0.205, whereas for the receptor, the Tm was 0.401, illustrating a confident structure prediction (threshold value lies between −2 and +2). However, both the models were further refined to improve the quality as well as the confidence score.

### 3.10. Protein Refinement of Vaccine and IKBKE Receptor

Out of the 5 output models predicted by the GalaxyRefine server, model 2 was selected for the vaccine construct and model 1 was finalized for the receptor based on the most improved Rama scores, the best RMSD values, and the least poor rotameters as shown in Figure 5. 

### 3.11. Vaccine Model Stability and Validation 

The refined 3D structure of the vaccine validated using the PROCHECK Ramachandran plot is shown in Figure 6A. According to the plot, 90.7% of the amino acids were in the core region, 8.5% were in the allowed region, 0.5% in general, and only 0.3% amino acids were plotted in the disallowed region, elucidating the structural stability of the vaccine.

### 3.12. Molecular Docking

As per the docking analysis predicted using Cluspro, a total of 10 models were analyzed. The best 9th model was selected based on the lowest energy and maximum members involved in the formation of the cluster, as shown in Figure 6B.

### 3.13. Molecular Dynamics Simulation 

The molecular simulation performed using iMODs, shown in Figure 7, interpreted the stability of the docked complex based on field force. It analyzed the deformability potential of the docked complex and predicted that the vaccine construct and IKBKE receptor interacted most stably at 500 ns, the calculated B-factor also supported a stable molecular interaction, the eigenvalues of 2.378923 × 10^−6^ illustrated the energy required to deform the complex with little variance. The covariance and elastic maps of the docked complex predicted little to no ambiguous interactions.

### 3.14. In-Silico Cloning with Snapgene

The SnapGene database was used to recover the sequence of the pET28(a) plasmid vector (shown in Figure 8A). The vaccine was cloned into the pET-28(a) vector by cutting the vector and vaccine construct with restriction enzymes *Sal1* and *EcoR1* at the MSC region for expression in Escherichia coli, as shown in Figure 8B. 

### 3.15. Immune Stimulation Analysis

C-immsim showed that the antigen count exponentially decreases until it reaches 0 in the first 5 days. The antibodies (IgG1, IgG2, and IgM) are induced in the disappearing period of the antigen and their count increases significantly in the first 15 days. After that, the quantity decreases slowly, becoming stable a month later. The individual and combined counts of the antigen and the antibodies are illustrated in Figure 8C.

## 4. Discussion

The Machupo virus may be aerosolized, food-borne, or spread directly. These viral particles come from the field mouse, *Calomys callosus*, and an infection reservoir [38]. Machupo flare-ups occur when abundant food supplies or human living and agricultural practices induce a rat population boom [7]. Machupo fever is hard to identify since it resembles Junin and Sabia, as well as dengue, malaria, and yellow fever [8]. Vaccination is beneficial for improving people’s health at a low cost and with proper help to inhibit the transmission of disorders worldwide [39]. The developmental process of vaccines requires significant expenditure of labor and a high price; however, immunoinformatic approaches reduce this burden. Presently, researchers are finding methods for producing and developing vaccines from the genomics or proteomics level of pathogens [40,41]. 

A Machupo virus vaccine has not been proposed yet because of the low severity and epidemiology of the disease; however, the world is always threatened by potential endemics and epidemics like the Bolivian Hemorrhagic fever; hence, there is a need to focus on its outer protein to contrive a preventative strategy. Notably, the targeted protein’s secondary structure and physicochemical properties render it highly antigenic for vaccine development [12]. This study, exploiting the NP’s nature has targeted the protein for the evaluation of potential T-cell and B-cell restricted epitopes. Similar methods have been utilized to form multi-epitope [13] and mRNA-based vaccines [42,43] for several other health issues recently.

B-cells were once the only source for vaccine development; however, computational biology now focuses on histocompatibility complex (MHC) T-cells with the most interacting human leukocyte antigen (HLA) scheme in a new clinical research field called computational biology [44]. To begin, we contrived the nucleoprotein’s linear and 3D structures. For the prediction of antigenicity in B-cell epitopes, two approaches from the IEDB database were used. The antigenicity score cutoff was established at 5.0 in this case as described earlier by Pyle and Whelan [45]. The prediction, identification, and processing of T-cells have been improving the vaccine potential since researchers have realized the T-cell-specific epitopes’ potential as viable vaccine candidates [30]. With an IC50 value of less than 100, our T-cell epitopes had strong reactivity as advocated previously by Yasmin et al. [46].

Potential vaccine protein candidates must be surface-exposed and immunologically recognized [47]. The selected nucleocapsid protein was chosen based on these parameters along with antigenicity and allergenicity analysis using VaxiJen and Aller-TOP v2 to offer clear evidence that they play a crucial role in pathogenicity, in line with previous studies [48]. Furthermore, to establish cellular and humoral immunity, both epitopes of B and T cells are united to generate a strong and sustained immune response, as done in the current study.

Utilizing both bioinformatics and immunologic approaches, it was concluded that the protein sequence designed does not carry allergenic or poisonous features. The study of Heo et al. [49] introduced a critical software, GalaxyRefine, for the purpose of refining vaccine constructs, which assisted in the improvement of the structural quality. The vaccine construct evaluated through the Ramachandran plot showed that most of the residues were present in the preferred region, and only a few were plotted in the disallowed region. The quality we predicted is an appropriate level of intendedness. Clustpro2.0 further demonstrated that model 9 of the vaccine-receptor complex showed the maximum interactions with the lowest energy values. The study of Ferron et al. [50] selected the required docked complex with the same characteristics. Following the acknowledged literature, the vaccine design in this study is predicted to successfully combat Bolivian Hemorrhage Fever across the world with a 99.99% coverage, albeit experimental proof-of-study will add considerably to the predicted vaccine candidate.

## 5. Conclusions

The current study was based on the effort revolving around an effective vaccine development against the Machupo Virus using computational analysis. The illustrated approach of in-silico analysis favors future experimental analysis as it saves cost and time taken to identify the target candidate and potential epitopes. For clinical trials, a validated computational approach, a strong immune response, and over 90% population coverage are deemed significant features, all of which are followed in the current study. The immune response predicted also indicated an effective elicitation with the antigen being excreted out of the host system within the first 5 days of injection. However, these results are mere predictions, even if validated, and to verify the vaccine’s efficacy, it must undergo experimental validation. Nonetheless, the design helps in eradicating the threat of the Machupo virus by putting forward the layout of an effective vaccine.

### Contribution/Prospects

The vaccine design against Bolivian Hemorrhage Fever is among the top priorities of the U.S. Department of Health and Human Services Public Health Emergency Medical Countermeasures Enterprise’s implementation plan.At present, there is no FDA-approved treatment for Bolivian Hemorrhage Fever (BHF). This computational vaccine model against the nucleocapsid protein of the Machupo virus has the capacity to assist in this regard.In previous research, the vaccine against viral glycoprotein of Machupo virus is proposed, but there was no particular focus on the nucleocapsid protein of the Machupo virus, which is the main protein interacting with the IKBKE receptor of the host cell.The vaccine design in this study using the reverse vaccinology approach will help researchers in wet lab-based development of a vaccine against Machupo virus in treating Bolivian Hemorrhage Fever.

## Figures and Tables

**Figure 1 vaccines-10-01732-f001:**
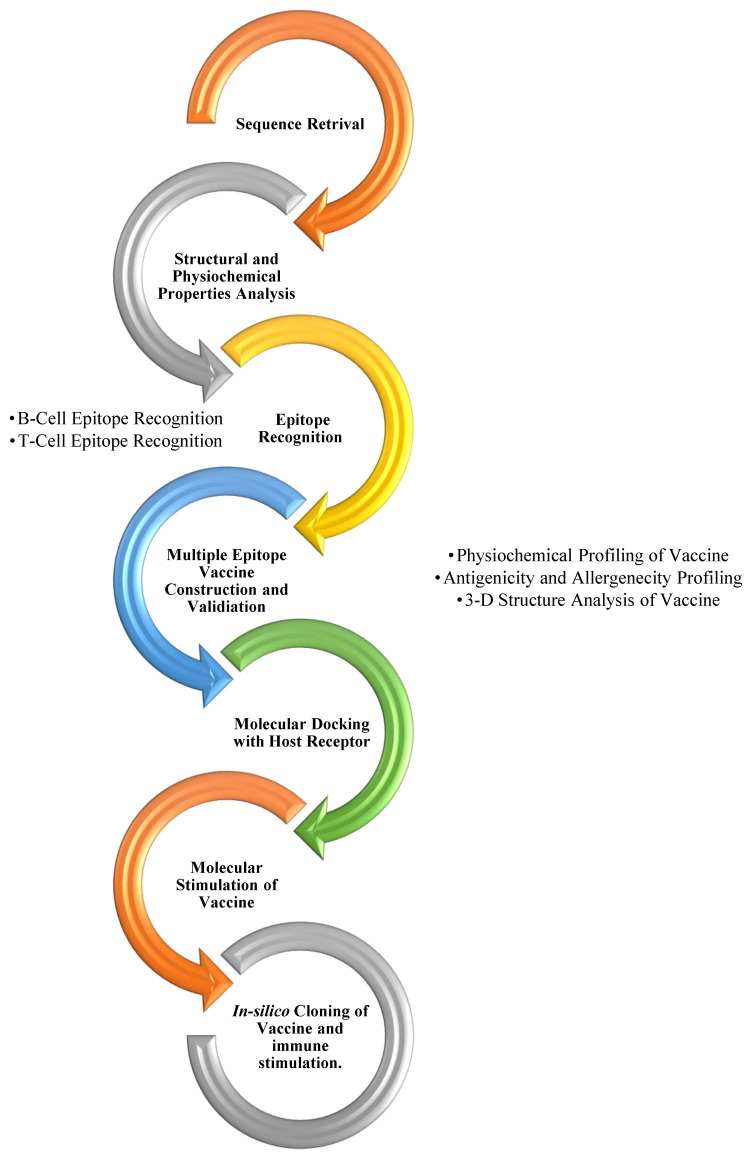
An overview of the process of designing a multiple epitope-based vaccine design against the Machupo virus.

**Figure 2 vaccines-10-01732-f002:**
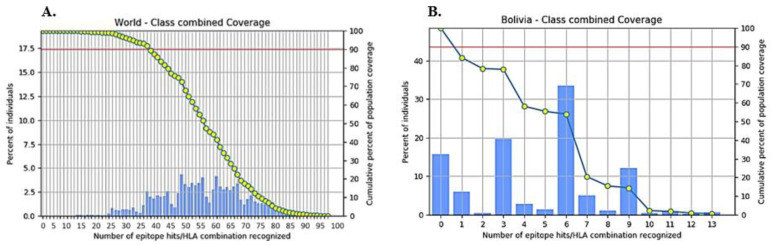
Population coverage of the potential epitopes and their target alleles. (**A**). World coverage; (**B**). Bolivian coverage.

**Figure 3 vaccines-10-01732-f003:**
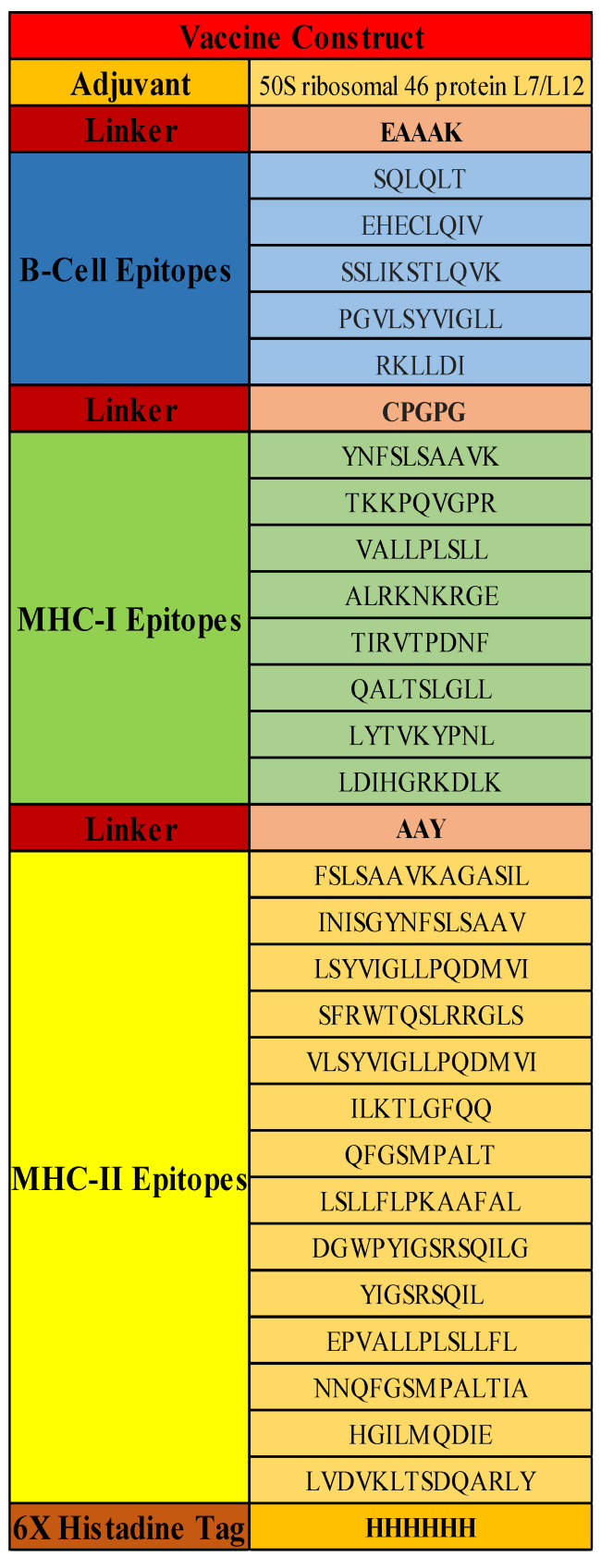
Vaccine Construct against the Nucleocapsid protein; 5 B-Cell Epitopes, 8 MHC-I epitopes, and 14 MHC-II epitopes.

**Figure 4 vaccines-10-01732-f004:**
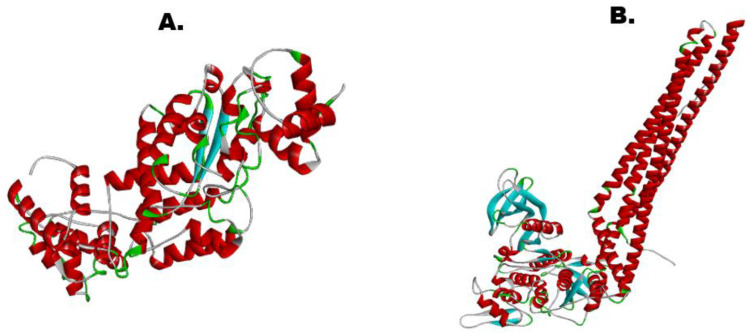
(**A**). 3-D structure of vaccine construct (Tm-Score: 0.205). (**B**). 3-D structure of IKBKE receptor (Tm-Score: 0.401).

**Figure 5 vaccines-10-01732-f005:**
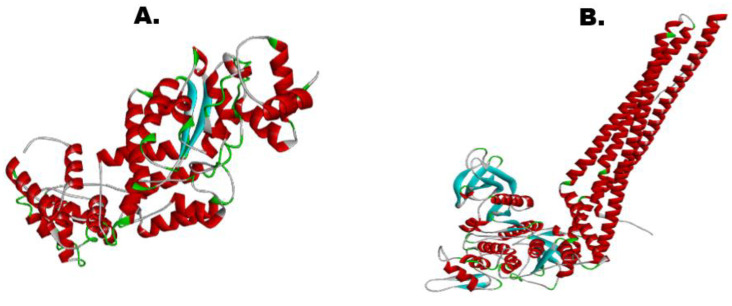
(**A**). Refined 3-D structure of vaccine candidate. (**B**). Refined 3-D structure of IKBKE receptor.

**Figure 6 vaccines-10-01732-f006:**
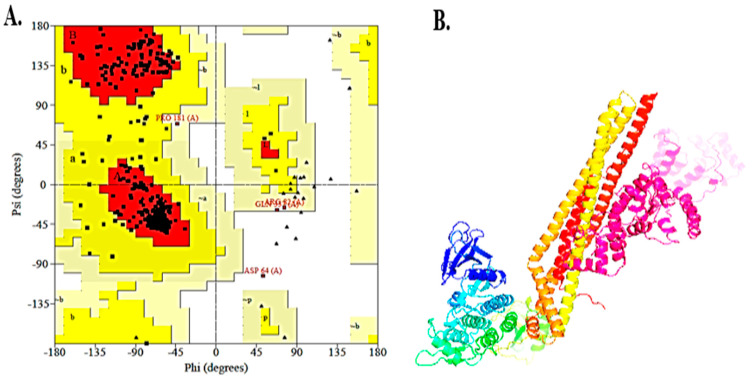
(**A**) Ramachandran Plot by PROCHECK Server; 90.7% of residues are in the core region, 8.5% lie in allowed zones and 0.3% are in disallowed region; (**B**) The docked complex via Cluspro, visualized on PyMol with a minimum energy score of −7.4 and 129 clusters. The colors indicate the different chains of the rector and the vaccine construct. The pink, orange, and yellow colors show the chain of the receptor molecule, whereas green, cyan, and blue indicate the vaccine molecule.

**Figure 7 vaccines-10-01732-f007:**
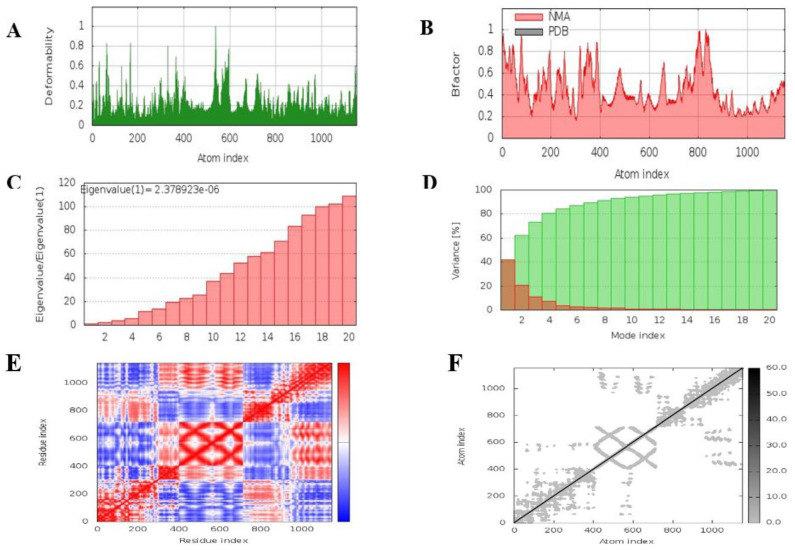
Molecular dynamics simulation of the docked complex. (**A**). Deformability of the complex; (**B**). B-Factor graph, the plot indicates a comparative PDB plot, but there are no validated structures of our molecules on PDB; therefore, the plot is blank (**C**). the eigenvalue plot illustrates the minimum energy required to deform the complex (**D**). Variance (**E**). Covariance Map correlated (red), uncorrelated (white), or anticorrelated (blue) motions (**F**). Elastic Network (grey matter indicates stiffer region).

**Figure 8 vaccines-10-01732-f008:**
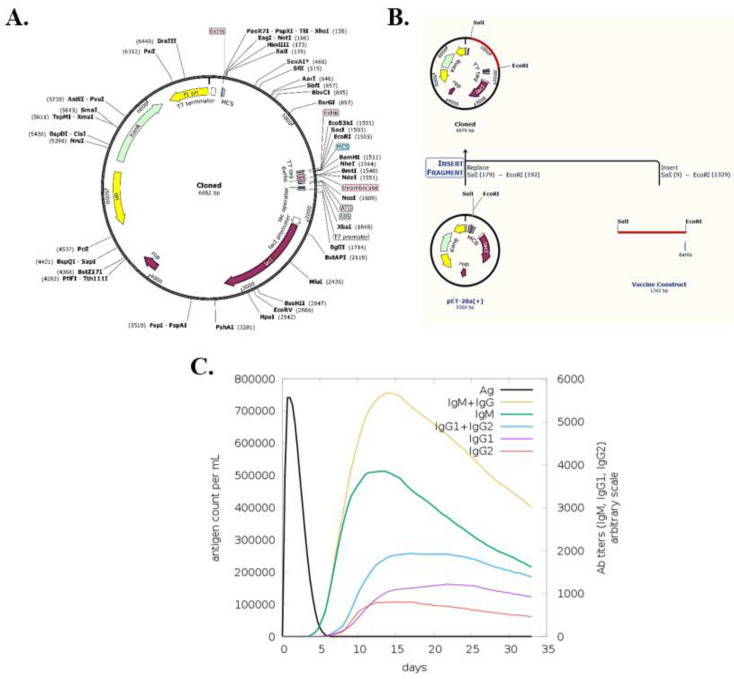
(**A**) The pEt28(a) vector; (**B**) the cloned vector with the red line showing our vaccine inside the plasmid; (**C**) the stimulation and response of IgM and IgG antibodies as compared to the antigen count over the course of the first 35 days.

**Table 1 vaccines-10-01732-t001:** Predicted B-cell epitopes based on antigenicity score.

No.	Start	End	Peptide	Length	Antigenicity Score
1	129	134	SQLQLT	6	1.4796
2	220	227	EHECLQIV	8	1.3906
3	271	281	SSLIKSTLQVK	11	0.6508
4	434	444	PGVLSYVIGLL	11	0.6472
5	459	464	RKLLDI	6	0.9124

**Table 2 vaccines-10-01732-t002:** Predicted MHC-I restricted epitopes computed from IEDB.

No.	Peptide	Alleles	Length	Antigenicity Score
1	YNFSLSAAVK	HLA-B*15:01, HLA-A*23:01, HLA-A*24:02, HLA-B*08:01, HLA-A*02:06, HLA-A*02:01, HLA-B*58:01, HLA-A*02:03, HLA-B*57:01, HLA-A*32:01, HLA-A*03:01, HLA-A*30:02, HLA-E*01:01, HLA-B*53:01, HLA-A*31:01, HLA-B*35:01, HLA-A*11:0, HLA-A*01:01, HLA-B*44:02, HLA-A*68:02, HLA-A*33:01, HLA-C*14:02, HLA-A*68:01, HLA-B*51:01, HLA-B*40:01, HLA-B*58:02, HLA-B*44:03, HLA-C*07:02, HLA-C*04:01, HLA-C*12:03, HLA-C*07:01, HLA-C*06:02, HLA-C*15:02, HLA-C*05:01, HLA-C*03:03	10	1.1965
2	TKKPQVGPR	HLA-A*02:06, HLA-C*08:02, HLA-C*04:01, HLA-A*02:01, HLA-C*15:02, HLA-B*40:01, HLA-A*23:01, HLA-B*35:01, HLA-A*68:02, HLA-A*01:01, HLA-B*15:01, HLA-B*51:01, HLA-A*02:03, HLA-C*05:01, HLA-B*58:01, HLA-B*53:01, HLA-A*30:01, HLA-B*44:02, HLA-B*58:02, HLA-B*57:01, HLA-A*26:01, HLA-B*07:02, HLA-A*24:02, HLA-E*01:01, HLA-A*33:01, HLA-A*30:02, HLA-A*31:01, HLA-B*44:03, HLA-A*11:01, HLA-A*03:01, HLA-A*32:01, HLA-A*68:01	10	1.6985
3	VALLPLSLL	HLA-A*23:01, HLA-B*44:02, HLA-A*30:02, HLA-A*24:02, HLA-B*15:01, HLA-B*44:03, HLA-A*02:06, HLA-A*26:01, HLA-B*07:02, HLA-A*02:01, HLA-B*08:01, HLA-B*57:01, HLA-A*01:01, HLA-A*68:02, HLA-B*58:01, HLA-B*40:01, HLA-C*14:02, HLA-A*32:01, HLA-B*53:01, HLA-E*01:01, HLA-B*58:02, HLA-C*04:01, HLA-B*35:01, HLA-B*51:01, HLA-C*07:01, HLA-C*07:02, HLA-C*06:02, HLA-C*12:03, HLA-C*15:02, HLA-C*08:02, HLA-C*05:01, HLA-C*03:03	9	1.0977
4	ALRKNKRGE	HLA-A*02:01, HLA-B*58:01, HLA-A*01:01, HLA-A*68:01, HLA-A*30:01, HLA-A*02:03, HLA-A*26:01, HLA-A*31:01, HLA-B*53:01, HLA-B*44:02, HLA-B*57:01, HLA-B*08:01, HLA-B*40:01, HLA-A*33:01, HLA-B*51:01, HLA-B*07:02	9	1.2132
5	TIRVTPDNF	HLA-B*08:01, HLA-A*02:06, HLA-A*02:01, HLA-B*15:01, HLA-B*53:01, HLA-A*23:01, HLA-A*24:02, HLA-B*35:01, HLA-B*51:01, HLA-A*68:02, HLA-B*40:01, HLA-A*30:01, HLA-A*01:01, HLA-A*26:01, HLA-B*58:01, HLA-B*57:01, HLA-B*44:02, HLA-A*31:01, HLA-A*33:01, HLA-A*30:02, HLA-A*32:01, HLA-A*03:01	9	0.8575
6	QALTSLGLL	HLA-A*02:03, HLA-A*24:02, HLA-A*30:01, HLA-B*58:01, HLA-B*08:01, HLA-B*53:01, HLA-A*32:01, HLA-A*02:01, HLA-B*35:01, HLA-A*23:01, HLA-B*44:02, HLA-B*57:01, HLA-A*30:02, HLA-A*01:01, HLA-B*07:02, HLA-A*03:01, HLA-A*11:01, HLA-A*31:01, HLA-B*40:01, HLA-A*33:01, HLA-B*44:03	9	0.8575
7	LYTVKYPNL	HLA-B*08:01, HLA-A*02:03, HLA-B*44:03, HLA-B*44:02, HLA-B*40:01, HLA-A*30:01, HLA-A*02:01, HLA-B*57:01, HLA-B*58:01, HLA-B*07:02, HLA-B*51:01, HLA-A*31:01	9	0.7742
8	LDIHGRKDLK	HLA-A*33:01, HLA-B*40:01, HLA-B*07:02, HLA-A*01:01, HLA-A*24:02, HLA-A*30:02, HLA-A*30:01, HLA-B*44:02, HLA-A*68:01, HLA-B*57:01, HLA-B*51:01, HLA-A*11:01, HLA-A*32:01, HLA-B*44:03	10	1.0902

**Table 3 vaccines-10-01732-t003:** Predicted MHC-II restricted epitopes computed from IEDB.

No.	Peptide	Alleles	Length	Antigenicity Score
1	FSLSAAVKAGASIL	HLA-DPA1*02:01/DPB1*14:01, HLA-DQA1*01:02/DQB1*06:02, HLA-DQA1*05:01/DQB1*03:01, HLA-DRB1*01:01, HLA-DRB1*04:01, HLA-DRB1*04:05, HLA-DRB1*07:01, HLA-DRB1*08:02, HLA-DRB1*09:01, HLA-DRB1*11:01, HLA-DRB1*12:01, HLA-DRB1*13:02, HLA-DRB1*15:01, HLA-DRB3*01:01, HLA-DRB3*02:02, HLA-DRB5*01:01	14	0.8104
2	INISGYNFSLSAAV	HLA-DPA1*01:03/DPB1*02:01, HLA-DPA1*02:01/DPB1*01:01, HLA-DPA1*02:01/DPB1*05:01, HLA-DPA1*03:01/DPB1*04:02, HLA-DQA1*01:01/DQB1*05:01, HLA-DQA1*05:01/DQB1*03:01, HLA-DRB1*01:01	14	0.6191
3	LSYVIGLLPQDMVI	HLA-DPA1*01:03/DPB1*06:01, HLA-DPA1*02:01/DPB1*01:01, HLA-DQA1*04:01/DQB1*04:02, HLA-DPA1*02:01/DPB1*05:01, HLA-DPA1*01:03/DPB1*06:01, HLA-DQA1*04:01/DQB1*04:02, HLA-DRB1*04:05, HLA-DRB4*01:01, HLA-DRB5*01:01, HLA-DRB1*10:01, HLA-DRB1*11:01, HLA-DRB1*04:01, HLA-DQA1*03:01/DQB1*03:01	14	0.7864
4	SFRWTQSLRRGLS	HLA-DQA1*02:01/DQB1*03:03, HLA-DQA1*02:01/DQB1*03:01, HLA-DQA1*05:01/DQB1*03:02, HLA-DQA1*05:01/DQB1*03:01, HLA-DQA1*02:01/DQB1*04:02, HLA-DQA1*04:01/DQB1*04:02, HLA-DQA1*05:01/DQB1*04:02, HLA-DQA1*03:01/DQB1*03:02	14	0.5757
5	VLSYVIGLLPQDMVI	HLA-DPA1*02:01/DPB1*05:01, HLA-DRB1*03:01, HLA-DRB1*04:01, HLA-DRB1*08:02, HLA-DRB1*11:01	14	0.5388
6	ILKTLGFQQ	HLA-DRB1*11:01, HLA-DRB1*08:02, HLA-DPA1*02:01/DPB1*05:01, HLA-DRB5*01:01, HLA-DPA1*03:01/DPB1*04:02, HLA-DPA1*02:01/DPB1*01:01, HLA-DRB4*01:01, HLA-DPA1*01:03/DPB1*02:01, HLA-DRB1*04:05, HLA-DRB1*04:01, HLA-DRB1*15:01, HLA-DRB1*07:01	9	0.1852
7	QFGSMPALT	HLA-DRB1*10:01, HLA-DRB1*08:01, HLA-DRB1*15:01, HLA-DRB1*04:01, HLA-DRB1*08:01, HLA-DRB1*07:01, HLA-DQA1*01:02/DQB1*05:01, HLA-DRB1*16:02, HLA-DRB1*04:05, HLA-DRB1*08:02, HLA-DRB1*12:01, HLA-DQA1*02:01/DQB1*03:01, HLA-DQA1*02:01/DQB1*03:03, HLA-DPA1*01:03/DPB1*04:02, HLA-DQA1*01:04/DQB1*05:03	9	0.8106
8	LSLLFLPKAAFAL	HLA-DRB1*03:01, HLA-DRB3*01:01, HLA-DRB1*04:01, HLA-DPA1*03:01/DPB1*04:02, HLA-DPA1*02:01/DPB1*01:01, HLA-DRB5*01:01, HLA-DRB4*01:01, HLA-DRB1*13:02	13	0.7809
9	DGWPYIGSRSQILG	HLA-DPA1*02:01/DPB1*01:01, HLA-DPA1*02:01/DPB1*05:01, HLA-DPA1*03:01/DPB1*04:02, HLA-DQA1*05:01/DQB1*03:01	14	0.7941
10	YIGSRSQIL	HLA-DPA1*02:01/DPB1*05:01, HLA-DRB1*08:02, HLA-DRB5*01:01, HLA-DRB4*01:01, HLA-DPA1*03:01/DPB1*04:02, HLA-DPA1*02:01/DPB1*01:01, HLA-DRB1*03:01, HLA-DPA1*01:03/DPB1*02:01, HLA-DRB1*12:01, HLA-DPA1*01:03/DPB1*04:01, HLA-DRB1*04:05	9	0.6841
11	EPVALLPLSLLFL	HLA-DPA1*01:03/DPB1*04:01, HLA-DPA1*01:03/DPB1*02:01, HLA-DRB4*01:01, HLA-DRB1*04:05, HLA-DRB1*12:01, HLA-DRB1*15:01, HLA-DRB1*04:01, HLA-DRB1*07:01, HLA-DRB1*01:01, HLA-DRB1*13:02, HLA-DRB1*09:01	14	0.9988
12	NNQFGSMPALTIA	HLA-DPA1*03:01/DPB1*04:02, HLA-DQA1*01:02/DQB1*06:02, HLA-DQA1*03:01/DQB1*03:02, HLA-DQA1*04:01/DQB1*04:02, HLA-DQA1*05:01/DQB1*03:01, HLA-DRB1*01:01, HLA-DRB1*03:01	14	0.7757
13	HGILMQDIE	HLA-DPA1*02:01/DPB1*01:01, HLA-DRB4*01:01, HLA-DPA1*01:03/DPB1*02:01, HLA-DQA1*03:01/DQB1*03:02, HLA-DRB1*04:05, HLA-DRB1*04:01, HLA-DRB1*15:01, HLA-DRB1*07:01	9	0.4930
14	LVDVKLTSDQARLY	HLA-DPA1*02:01/DPB1*14:01, HLA-DQA1*01:02/DQB1*06:02, HLA-DQA1*05:01/DQB1*03:01, HLA-DRB1*01:01, HLA-DRB1*03:01, HLA-DRB1*04:01, HLA-DRB1*07:01, HLA-DRB1*08:02, HLA-DRB1*09:01	14	0.5489

## Data Availability

All major data generated and analyzed in this study are included in this manuscript.

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
