# Peer review of "Immunoinformatics Approach to Design Multi-Epitope-Based Vaccine against Machupo Virus Taking Viral Nucleocapsid as a Potential Candidate"

_vaccines, 2022, doi:10.3390/vaccines10101732_

Round 1

Reviewer 1 Report

            The authors designed by immune-informatics, a vaccine composed of multiple epitopes of the nucleoprotein Machupo virus, as a potential candidate to protect against this viral disease. Some aspects should be considered before publication.

Major comments:

1.      The main limitation which is not addressed by the authors is the rationale of targeting the NP protein as immunogen for inducing a protective immune response, while it is expected and known that neutralizing antibodies are raised against the GP, the viral protein interacting with the cellular receptor.

2.      The authors should modulate their prediction on rapid induction and long-lasting antibody response (see Abstract).

Minor comments:

3.      Introduction: page 2 line 60: viruses are not individuals.

4.      Introduction, page 2 line 88: please insert a coma between dengue and malaria

5.      Introduction, page 2 lines 91-96: it is not clear if the authors are talking about the NP protein or the GP one. As stated before, the Introduction should include argumentation on why the NP protein was chosen as a target for inducing immune protective response. Some anecdotal cases reflecting the high mortality rate of this virus could be eliminated. The introduction instead should include the rationale of this study, based on published previous studies. Some previous reports have studied indeed the arenaviruses NP as immunogen, but they are not cited in this manuscript.

6.      The authors should mention that an attenuated vaccine has been developed already for a Machupo virus relative, the Argentinian Hemorrhagic virus.

7.      The Contribution/Prospects mentioned by the authors should be focused on their results.

Author Response

Major comments:

  1. The main limitation which is not addressed by the authors is the rationale of targeting the NP protein as immunogen for inducing a protective immune response, while it is expected and known that neutralizing antibodies are raised against the GP, the viral protein interacting with the cellular receptor.

AR: Thank you for your comment. The rationale for using NP instead of GP was based on the fact that the generation of neutralizing antibodies against GPs has been extensively studied. Furthermore, the NP contains the GP segment, therefore, we chose the NP as a potential target to develop an additional or alternative therapy. The manuscript has been updated [Lines 103-115].

  1. The authors should modulate their prediction on rapid induction and long-lasting antibody response (see Abstract).

AR: Thank you for your comment. We have modulated the predictions please see revised manuscript [abstract, lines 59-60], section 3.14 and figure 8C.

Minor comments:

  1. Introduction: page 2 line 60: viruses are not individuals.

AR: Thank You for your comment. It has been replaced please see revised manuscript [Line 67].

  1. Introduction, page 2 line 88: please insert a coma between dengue and malaria.

AR: Thank You for your comment. Coma inserted, please see revised manuscript  [Line 95]

  1. Introduction, page 2 lines 91-96: it is not clear if the authors are talking about the NP protein or the GP one. As stated before, the Introduction should include argumentation on why the NP protein was chosen as a target for inducing immune protective response. Some anecdotal cases reflecting the high mortality rate of this virus could be eliminated. The introduction instead should include the rationale of this study, based on published previous studies. Some previous reports have studied indeed the arenaviruses NP as immunogen, but they are not cited in this manuscript.

AR: Thank You for your comment. We are talking about NP protein, which is transcribed by 1). S segment and 2). Viral glycoprotein. The rationale for choosing NP protein considering previous studies has been updated. Please see revised manuscript [Lines 103-115].

  1. The authors should mention that an attenuated vaccine has been developed already for a Machupo virus relative, the Argentinian Hemorrhagic virus.

AR: Thank you for your comment. Manuscript updated, please see revised manuscript [Lines 119-122].

  1. The Contribution/Prospects mentioned by the authors should be focused on their results.

AR: Thank You for your comment. The 2nd and 3rd prospects reflect the study’s results. please see revised manuscript, lines 478-482.

Reviewer 2 Report

The manuscript entitled “Immunoinformatics Approach to Design Multi-Epitope-Based Vaccine against Machupo Virus Taking Viral Nucleocapsid as a Potential Candidate” is a computational work based on reverse vaccinology to design a subunit vaccine. This work has several flaws that need to be revised to make it publishable. My comments are mentioned below

1.      Abstract: This part is written well and reflects the work done in this manuscript.

2.      Introduction

·        Line 63: Universe should be replaced with world

3.      Methodology

·        Authors should remove the dated of access for example,“accessed on 20th July 2022” for all tools. It does not make any scientific contribution.

·        Line 122: Please clarify the reason for selecting Bepipred out of seven methods available to predict the B-cell epitope from the IEDB server

·        Line 129: Author should rewrite this sentence as it seems confusing “and for the MHC-II-specific epitopes prediction, IEDB MHC-I tool was used”. I think authors should replace MHC-II with MHC-I.

·        Line 128: Mention the full form of “CD-based”

·        Line 133: Authors should specify which allele they chose to predict MHC-I and MHC-II epitope prediction and why?

·        Line 141: Rewrite “all 8 of MHC- and 12 out of 14 epitopes of MHC-II with their target alleles were taken as input” it should be all 8 of MHC-I and 12 out of 14 epitopes of MHC-141 II with their target alleles were taken as input.

4.      Results

·        Line 204: Authors should elaborate what they want to conclude with the obtained value of instability index, aliphatic index and GRAVY. As per their data, the instability index value is 42.31, which means protein is unstable. How would you justify it?

·        Line 213-214: “The maximum and minimum antigenicity were 1.215 and 0.860, respectively, with the average antigenicity being 1.215.” Authors should recheck the maximum and average values.

·        Line 220: Authors should mention the range of IC50 value and their criteria for selecting epitopes with IC50 value less than 100.

·        Table-2: As per the IEDB MHC-II epitope prediction server, the lowest epitope length is 11 amino acids; then how did the authors predict Th cell epitopes of 9 amino acids?

·        Authors should clearly mention the alleles associated with the selected epitopes and allele-specific population coverage. Authors should also elaborate why they selected a specific allele.

·        Authors should separately describe MHC-I and MHC-II epitope prediction results

·        Line 229: “87.31% for the selected MHC-II epitopes” authors should mention which countries were not covered with these MHC-II epitopes as the coverage is only 87.31%. Are that countries endemic to this disease?

·        Line 228-229: In the introduction, the authors mentioned that this disease isn't spread worldwide, then why do they predict the population coverage for the worldwide population, it will be nice to check the coverage of the predicted epitope in the endemic countries.

·        Line 239: Authors should mention the role of linkers when used for different epitopes.

·        Authors should cite all the linkers source publication

·        Line 240: There is a mistake in the selection of the linker, I think its the GPGPG linker but not the CPGPG linker. If there is certain literature that uses CPGPG linker, author should cite it.

·        Line 257: “construct as hydrophobic” then how will it be administered?

·        Line 260-262: Cite this sentence

·        Figure 6: Resolution should be increased to make it clearly visible

·        Line 294: Authors should mention the plot for RMSD and RMSF. Simultaneously describe figure 7 and its all components so that anyone can understand.

5.      Discussion

·        Poorly discussed with irrelevant citation

6.      References

·        There are several self-citation that actually do not correlate with the subject, they must be replaced with relevant citations.

Author Response

  1. Abstract: This part is written well and reflects the work done in this manuscript.

AR: Thank you for acknowledging it. Your appreciation means a lot to us.

  1. Introduction
  • Line 63: Universe should be replaced with world

AR: Thank You for your comment. It has been replaced. Please see revised manuscript [Line: 70]

  1. Methodology
  • Authors should remove the dated of access for example,“accessed on 20th July 2022” for all tools. It does not make any scientific contribution.

AR: Thank You for your comment. It has been removed.

  • Line 122: Please clarify the reason for selecting Bepipred out of seven methods available to predict the B-cell epitope from the IEDB server

AR: Thank You for your comment. BepiPred uses a hidden Markov model along with the propensity scale approach to estimate the location of linear B-cell epitopes. Please see revised manuscript [Lines: 153-155].

  • Line 129: Author should rewrite this sentence as it seems confusing “and for the MHC-II-specific epitopes prediction, IEDB MHC-I tool was used”. I think authors should replace MHC-II with MHC-I.

AR: Thank You for your comment. IEDB (Immune Epitope Database) was accessed at http://tools.iedb.org/mhcii/ for the prediction of CD-based helper T cell epitopes, and the IEDB MHC-I tool was utilized for the prediction of MHC-I-specific epitopes. Rewrote and replaced MHC-II with MHC-I. Please see revised manuscript [Lines 158-160] 

  • Line 128: Mention the full form of “CD-based”

AR: Thank You for your comment. CD-Based “Conserved Domain Based”, full form written in the manuscript. Please see revised manuscript [Line 159]

  • Line 133: Authors should specify which allele they chose to predict MHC-I and MHC-II epitope prediction and why?

AR: Thank You for your comment. Information has been added to the revised manuscript. Please see revised manuscript [Lines 163-164].

  • Line 141: Rewrite “all 8 of MHC-and 12 out of 14 epitopes of MHC-II with their target alleles were taken as input” it should be all 8 of MHC-I and 12 out of 14 epitopes of MHC-141 II with their target alleles were taken as input.

AR: Thank You for your comment. It has been replaced. Please see revised manuscript [Line 172-173]

  1. Results
  • Line 204: Authors should elaborate what they want to conclude with the obtained value of instability index, aliphatic index and GRAVY. As per their data, the instability index value is 42.31, which means protein is unstable. How would you justify it?

AR: Thank You for your comment. This is the instability index of the protein retrieved from the NCBI. The instability index of our constructed vaccine is 35.13. In our study we are concerned with the physicochemical properties of our construct and not with the retrieved protein. The physicochemical properties of the retrieved protein were analyzed just for the general information of the protein.

  • Line 213-214: “The maximum and minimum antigenicity were 1.215 and 0.860, respectively, with the average antigenicity being 1.215.” Authors should recheck the maximum and average values.

AR: Thank You for your comment. The maximum and minimum antigenicity were 1.4796 and 0.6472, respectively, with the average antigenicity being 1.015. Rechecked and corrected. Please see revised manuscript [Line 261-263].

  • Line 220: Authors should mention the range of IC50 value and their criteria for selecting epitopes with IC50 value less than 100.

AR: Thank You for your comment. The range and criterion have been updated in the manuscript. Please see revised manuscript [Lines 271-274]

  • Table-2: As per the IEDB MHC-II epitope prediction server, the lowest epitope length is 11 amino acids; then how did the authors predict Th cell epitopes of 9 amino acids?

AR:  Thank You for your comment. The results of MHC-II-specific epitopes provide two columns, one belonging to the core epitopes (AA length 9) and the other one with the expanded epitopes (variable lengths). The four epitopes with 9 AA lengths were selected because these core epitopes provided multiple expanded epitopes with lower IC50 values and similar allele restrictions. To avoid redundancy, we used the core epitopes instead of the expanded ones. The other epitopes were chosen from the expanded epitopes’ column.

  • Authors should clearly mention the alleles associated with the selected epitopes and allele-specific population coverage. Authors should also elaborate on why they selected a specific allele.

AR: Thank You for your comment. The alleles associated with the specific epitopes are updated in tables 2 and 3. Allele-specific population coverage is not discussed because it does not add significant information to the study. The authors did not select a specific allele. All the alleles that restricted the finalized epitopes at an IC50 value under 100 were chosen for the population coverage analysis. Manuscript has been revised and updated. Please see revised manuscript [Lines 289-291]

  • Authors should separately describe MHC-I and MHC-II epitope prediction results.

AR: Thank You for your comment. The adjustments have been made accordingly. Please see revised manuscript [Lines 277-283]

  • Line 229: “87.31% for the selected MHC-II epitopes” authors should mention which countries were not covered with these MHC-II epitopes as the coverage is only 87.31%. Are that countries endemic to this disease?

AR: Thank you for your comment. The population coverage percentage for epitopes does not relate to countries, instead it defines the number of people in the selected region that carry the alleles restricting a specific epitope. In other words, we examined the number of people who carry the alleles restricting the selected epitopes for the vaccine design with an IC50 value lower than 100. The percentages (for instance, 87.31% for MHC-II epitopes) demonstrate that 87.31% people of the selected region (in our case, the world) carry the MHC-II allele that will restrict and present the epitopes to the immune cells, hence providing the vaccine efficacy of 87.31% for the MHC-II alleles.

  • Line 228-229: In the introduction, the authors mentionedthat this disease isn't spread worldwide, then why do they predict the population coverage for the worldwide population, it will be nice to check the coverage of the predicted epitope in the endemic countries.

AR: Thank You for your comment. The purpose of this study was to prevent a potential epidemic caused by Machupo virus, which is why the world population was selected to predict the vaccine’s efficacy. However, acknowledging the need of endemic country population coverage, we have added the Bolivian coverage in figure 2B and lines 294-295. Please see revised manuscript.

  • Line 239: Authors should mention the role of linkers when used for different epitopes.

AR: Thank You for your comment. The role of linkers has been added. Please see revised manuscript [Lines 307-308, 311-315]

  • Authors should cite all the linkers source publication

AR: Thank You for your comment. The source publication has been added. Please see revised manuscript [Line 307].

  • Line 240: There is a mistake in the selection of the linker, I think its the GPGPG linker but not the CPGPG linker. If there is certain literature that uses CPGPG linker, author should cite it.

AR: Thank You for your comment. We apologize for the random error. CPGPG is replaced with GPGPG. Please see revised manuscript [Lines 307 and 313]

  • Line 257: “construct as hydrophobic” then how will it be administered?

AR: Thank You for your comment. According to RaptorX, the construct is 31% hydrophilic and according to GRAVY, the construct is slightly hydrophobic. This slight hydrophobicity can be dealt with in-vitro using hydrophilic helping linkers and adjuvants.

  • Line 260-262: Cite this sentence

AR: Thank You so much. Sentence cited.

  • Figure 6: Resolution should be increased to make it clearly visible.

AR: Thank You for your comment. Resolution improved.

  • Line 294: Authors should mention the plot for RMSD and RMSF. Simultaneously describe figure 7 and its all components so that anyone can understand.

AR: Thank You for your comment. None of the tools available freely provided the plots for RMSD or RMSF. Furthermore, every component of figure 7 is provided in the figure legend and described in section 3.13.

  1. Discussion
  • Poorly discussed with irrelevant citation

AR: Thank You for your comment. Discussion has been replaced.

  1. References
  • There are several self-citation that actually do not correlate with the subject, they must be replaced with relevant citations.

AR: Thank You for your comment. The study has reported five self-citations.

  1. The first two references are examples of computational vaccine studies evidencing prodigious advancements in the relevant field.
  2. We have used the figure template of the third reference.
  3. The last two references relate to the advancements of computational vaccine studies and mRNA vaccine designs, respectively. However, both have been supplemented with other citations.

Round 2

Reviewer 1 Report

The authors addressed the concerns of the reviewers.

Author Response

The authors addressed the concerns of the reviewers.

AR: Thank you very much for your response. We really appreciate it.

Reviewer 2 Report

The authors have significantly improved on the paper and have addressed all previously mentioned concerns. However, still need a few corrections.

1. Author should clearly define the MHC-II epitope selection criteria in the result sections as like they explained in the rebuttal letter. They should mention in the manuscript  that they selected core residues (9mer) and whole epitope (12-18 mers). 

2. Authors should cite each and every tool that they used in methods just after the URL.

3. I will strongly recommend to remove inappropriate self-citations (Ref no: 13, 14, 17, 24)

4. Line 402: I will recommend adding a few more references from other authors. A few suggestions are mentioned below

Vaccine 36 (30), 4555-4565; 

Vaccines 20219(6), 669

Journal of Biomolecular Structure and Dynamics 37 (9), 2381-2393; 

International journal of biological macromolecules 122, 1203-1211

Author Response

The authors have significantly improved on the paper and have addressed all previously mentioned concerns. However, still need a few corrections.

  1. Author should clearly define the MHC-II epitope selection criteria in the result sections as like they explained in the rebuttal letter. They should mention in the manuscript that they selected core residues (9mer) and whole epitope (12-18 mers). 

AR: Thank you so much for your response. We have updated the manuscript [Lines 264-270].

  1. Authors should cite each and every tool that they used in methods just after the URL.

AR: Thank you so much for your response. Citations have been added accordingly.

  1. I will strongly recommend to remove inappropriate self-citations (Ref no: 13, 14, 17, 24)

AR: Thank you so much for your response. Self-citations have been removed.

  1. Line 402: I will recommend adding a few more references from other authors. A few suggestions are mentioned below

Vaccine 36 (30), 4555-4565; 

Vaccines 20219(6), 669

Journal of Biomolecular Structure and Dynamics 37 (9), 2381-2393; 

International journal of biological macromolecules 122, 1203-1211

AR: Thank you so much for your response. The manuscript has been updated and more references have been added. Please see revised manuscript.